# Data Augmentation Techniques for Machine Learning Applied to Optical Spectroscopy Datasets in Agrifood Applications: A Comprehensive Review

**DOI:** 10.3390/s23208562

**Published:** 2023-10-18

**Authors:** Ander Gracia Moisés, Ignacio Vitoria Pascual, José Javier Imas González, Carlos Ruiz Zamarreño

**Affiliations:** 1Department of Electrical, Electronic and Communications Engineering, Public University of Navarra, Campus Arrosadía, 31006 Pamplona, NA, Spain; ignacio.vitoria@unavarra.es (I.V.P.); josejavier.imas@unavarra.es (J.J.I.G.); carlos.ruiz@unavarra.es (C.R.Z.); 2Pyroistech S.L., C/Tajonar 22, 31006 Pamplona, NA, Spain; 3Institute of Smart Cities, Public University of Navarra, Campus Arrosadía, 31006 Pamplona, NA, Spain

**Keywords:** optical spectroscopy, agrifood industry, artificial intelligence, data augmentation (DA), generative adversarial networks (GANs)

## Abstract

Machine learning (ML) and deep learning (DL) have achieved great success in different tasks. These include computer vision, image segmentation, natural language processing, predicting classification, evaluating time series, and predicting values based on a series of variables. As artificial intelligence progresses, new techniques are being applied to areas like optical spectroscopy and its uses in specific fields, such as the agrifood industry. The performance of ML and DL techniques generally improves with the amount of data available. However, it is not always possible to obtain all the necessary data for creating a robust dataset. In the particular case of agrifood applications, dataset collection is generally constrained to specific periods. Weather conditions can also reduce the possibility to cover the entire range of classifications with the consequent generation of imbalanced datasets. To address this issue, data augmentation (DA) techniques are employed to expand the dataset by adding slightly modified copies of existing data. This leads to a dataset that includes values from laboratory tests, as well as a collection of synthetic data based on the real data. This review work will present the application of DA techniques to optical spectroscopy datasets obtained from real agrifood industry applications. The reviewed methods will describe the use of simple DA techniques, such as duplicating samples with slight changes, as well as the utilization of more complex algorithms based on deep learning generative adversarial networks (GANs), and semi-supervised generative adversarial networks (SGANs).

## 1. Introduction

Optical spectroscopy is the discipline that covers how light interacts with matter [1]. Powerful devices rely on optical spectroscopy for in situ sample measurement, providing optimal results quickly and at a reduced cost in comparison to standard laboratory tests, and without damaging or degrading the sample [2,3]. When light interacts with a sample, part of it can be reflected, absorbed, or transmitted, depending on the sample’s chemical composition and physical properties. A simple schematic representation of this phenomenon is shown in Figure 1.

Analyzing the interaction of the incident light with the sample allows us to determine the properties of the sample [4]. Optical spectroscopy, specifically, measures the intensity of the radiation at various wavelengths. In this study case, the electromagnetic radiation in the ultraviolet (UV), visible (VIS), and infrared (IR) ranges is considered. Reflected radiation typically contains information about the sample’s surface, while transmitted radiation provides an insight into some properties of the bulk material, allowing, for example, the quality of products to be determined [5]. The absorption of radiation influences the reflected spectrum, shaping it further, which aids in the study of mineral color [6]. Consequently, optical spectroscopy is commonly used for mineral analysis, food quality assessment, chemical reaction monitoring [7], temperature measurement [8], and impurity detection [9].

In recent times, there has been a growing emphasis on ensuring the quality and safety of food products as well as production processes, making necessary a thorough analytical characterization of food, alongside the utilization of online detection techniques. UV–VIS–NIR spectroscopy, being a rapid, non-destructive method, is increasingly being employed for the evaluation of agricultural products and food quality in agrifood industry chains, with application across a diverse variety of products and fields, as presented in reviews on this topic [10,11,12,13]. For instance, VIS–NIR spectroscopy has been applied in the food chain to determine the chemical composition of chicken breast and thigh muscles [14]. In another instance, optical spectroscopy has been utilized in the agrifood chain of olive oil, a crucial aspect of the Mediterranean diet, which is known for being low in saturated fats due to the use of olive oil. Emerging techniques are now being developed to determine the quality and authenticity of olive oil [15], or to determine characteristics such as acidity, moisture, or peroxide levels in olive oil [16].

The methods proposed are not limited to specific products, they can also be employed to evaluate chemical compounds or molecules in different substances. A case in point is the assessment of formaldehyde in the agrifood chain, as presented in review [17]. Formaldehyde, a colorless, highly volatile, and flammable gas at normal temperature and pressure, has significant health implications due to its toxicological properties. Presently, the regulatory organization have imposed legal pressure to control the production of transgenic products, further accentuating the importance of reliable analytical methods in ensuring food safety and compliance, as presented in [18].

Machine learning and deep learning are both subfields of artificial intelligence. While machine learning employs algorithms that learn patterns from data and improve their performance over time, deep learning relies on neural networks with multiple layers, allowing more complex patterns and representations to be learned [19], and has proved to be especially effective in dealing with large, high-dimensional data [20]. Machine learning and deep learning algorithms have been widely used to develop a lot of new kinds of methods for analyzing spectroscopic data obtained from agrifood processes [21,22,23]. These methods are able to automatically identify some patterns and correlations between the data that would be difficult, or impossible, for humans to find without them.

However, obtaining accurate measurements may not always be easy or repeatable, and the process of collecting a good batch of spectroscopy data can be tedious, time consuming, and expensive [24,25,26], especially when specialized machines or chemical compounds are required [27]. Since machine learning and deep learning models rely heavily on the availability of large and diverse datasets for accurate and effective training, the lack of sufficient data in optical spectroscopy datasets is a relevant problem. Insufficient data can lead to overfitting, where the model learns the noise in the training data instead of the underlying patterns, resulting in poor generalization with new data [28]. Moreover, the scarcity of data can hinder the model’s ability to capture complex relationships between variables, ultimately affecting its performance and predictive capabilities [29]. In cases where the dataset is imbalanced, models tend to be biased towards the majority class, leading to suboptimal predictions for the minority class [30].

Data augmentation (DA) techniques can help to alleviate these issues by artificially increasing the size and diversity of the dataset, thus enhancing model performance and generalizability [31]. DA encompasses all the techniques employed to expand the number of samples in a dataset [32]. DA makes the training process more complex, but with the advantage of obtaining a more robust model with higher accuracy than one without the use of DA [33,34]. Moreover, DA techniques can help to reduce the costs and complexities of optical spectroscopy data collection [35] and it is common to find applications that include these tools for synthetic data generation.

Unlike previous reviews that may not have explored this particular intersection of technologies, our review sheds light on the potential of leveraging advanced machine learning techniques to significantly improve data augmentation in agrifood optical spectroscopy. Additionally, our manuscript provides a meticulous review of recent publications, offering a comprehensive and systematic analysis of the current state of data augmentation techniques in this field.

The text is organized as follows. The second section (Section 2) is focused on how computers are able to learn from data, departing for a brief explanation of the backpropagation algorithm, followed by a presentation of the most typical metrics employed to evaluate the performance of machine learning algorithms, from regression and classification tasks. The next section is focused on DA techniques (Section 3) and their ability to enhance datasets, and their applications in the agrifood industry. The first part of the section covers non-DL DA methods, typically based on algorithms and techniques that employ random noise to synthesize new samples. The last part of the section is focused on DA methods that are related to DL algorithms, specifically, on those ones that are based on a GAN architecture, and semi-supervised GAN algorithms, highlighting their potential for enhancing datasets in the field of optical spectroscopy applied to the agrifood industry. Finally, some remarks and conclusions are included at the end of this work.

## 2. A Brief Introduction to AI: Training and Evaluation

Machine learning is the science of programming computers to learn from data. A general way to express this is to say that in employing ML algorithms, computers obtain the ability to learn from data without being programmed. Another way to express it was proposed by Tom Mitchell in 1997, “A computer program is said to learn from experience (E) with respect to some class of tasks (T) and performance measure (P), if its performance at tasks (T) measured by P, improves with experience (E)”.

Given these two previous definitions, this section will summarize the main algorithms that allow computers to train by themselves, and various methods for evaluating their performance on different tasks.

### 2.1. Backpropagation in Training

Methods based on machine learning or deep learning require a prior training process. This process can be explained using the backpropagation method. Backpropagation (BP) was proposed by David Rumelhart, Geofrey Hinto, and Ronald Williams in 1986, introducing the backpropagation algorithm, which is still in use today [36]. In essence, it is a gradient descent algorithm that efficiently computes gradients automatically in just two steps: for each training instance, the backpropagation algorithm first makes a prediction (forward step) and measures the error. It then goes back through each layer in reverse order to measure the error contribution of each connection (backward step), and finally adjusts the connection weights to reduce the error (gradient descent step) [37].

The backpropagation process is further detailed in the points below:It takes a mini-batch from the training dataset.It sends the mini-batch though the network’s input layer, on to the first hidden layer, and computes the outputs of all neurons in this layer. This result is then passed to the next layer. This process continues until it reaches the output layer. This is the forward pass, similar to making predictions, but intermediate values are kept for calculations.A loss function, which compares the desired output with the actual output of the network, is used, and it returns an error metric.It calculates how much each output connection contributes to the error. It measures how many error contributions come from each connection of the previous layer. This process is performed all the way back to the input layer.The gradient descent process is applied to adjust the weights of the network using the error gradients that were just calculated.

Gradient descent is a generic optimization algorithm capable of finding optimal solutions to a wide range of problems [38]. The general idea of gradient descent is to iteratively adjust parameters in order to minimize the loss function. To achieve this, it measures the local gradient of the error (ERR) function with respect to the parameter vector and goes in the direction of the descending gradient. It first calculates how much the loss function changes with respect to parameter *ϴ_j_* if *ϴ_j_* changes slightly—in other words, it computes the partial derivative of the loss function with respect to the parameter *ϴ_j_*. This is shown in Equation (1):(1)$$\frac{\partial }{\partial \theta_{j}}ERR\left(\theta \right)$$

Instead of calculating the partial derivatives for each parameter, we can compute the gradient vector for all parameters, defined as in Equation (2):(2)$$\nabla_{\theta }ERR=\left[\begin{array}{c} \frac{\partial }{\partial \theta_{0}}ERR\left(\theta \right)\\ \begin{array}{c} \frac{\partial }{\partial \theta_{1}}ERR\left(\theta \right)\\ \ldots \end{array}\\ \frac{\partial }{\partial \theta_{n}}ERR\left(\theta \right) \end{array}\right]$$

Once the direction of the gradient vector is known, all that is needed is to move in the opposite direction to achieve gradient descent. This is where the learning rate (*lr*) comes into play, which is a hyperparameter that, when applied, reduces the impact of the gradient vector. The gradient descent is formulated as in Equation (3):(3)$$\theta^{next\;step}=\theta -lr*\;\nabla_{\theta }ERR\left(\theta \right)$$

More information about gradient descent can be found in [39], where the author explains different variants of gradient descent and compares all of them in terms of their training speed, convergence rate, performance, and pros and cons.

### 2.2. Model Performance Evaluation

One important task in building any ML or DL algorithm is to evaluate its performance. The primary goal of this step is to assess how effectively the model learns and makes predictions from given data, and to ascertain its ability to generalize to unseen data. Different metrics are applied for the tasks of regression and classification, but there are also some metrics that can be applied for both tasks. In this sense, we could optimize a model to perform better for certain outcomes and, therefore, we might use different metrics to select the final model to use. Because of this tradeoff, there is a variety of metrics to use in every specific case.

#### 2.2.1. Metrics for Regression Models

The evaluation of ML and DL algorithms, especially in the context of regression analysis, relies on statistical metrics such as the mean absolute error (*MAE*), mean squared error (*MSE*), root mean squared error (*RMSE*), and coefficient of determination (*R*^2^) [40].

The *MAE* represents the average of the absolute difference between the estimates and the true values in the dataset. *MAE* aligns with the expected loss L1, and it is formulated as in Equation (4):(4)$$\;MAE=\frac{1}{N}\overset{N}{\underset{n=1}{\sum }}\left|y_{n}-{\hat{y}}_{n}\right|$$

Similarly, *MSE* represents the average of the squared difference between the estimated and true values. It corresponds to the expected loss for L2, and it is formulated as in Equation (5):(5)$$\;MSE=\frac{1}{N}\overset{N}{\underset{n=1}{\sum }}{\left(y_{n}-{\hat{y}}_{n}\right)}^{2}$$

*RMSE* is the square root of *MSE*, and it provides a measure of the standard deviation of residuals. The *RMSE* describes how well the ML or DL model can predict the estimates of a response variable, and it is defined in Equation (6):(6)$$\;RMSE=\sqrt{MSE}=\sqrt{\frac{1}{N}\overset{N}{\underset{n=1}{\sum }}{\left(y_{n}-{\hat{y}}_{n}\right)}^{2}}$$

*R*^2^, a scale-free score, is a metric that represents the proportion of the variance in the dependent variable attributed to the ML or DL model. Irrespective of the values being small or large, the value of *R*^2^ will be less than one, indicative of the model’s effectiveness in explaining the variability in the response variable. *R*^2^ is formulated as in Equation (7):(7)$$R^{2}=1-\frac{\sum {\left(y_{n}-{\hat{y}}_{n}\right)}^{2}}{\sum {\left(y_{n}-\bar{y}\right)}^{2}}$$

Both *MSE* and *RMSE* amplify large prediction errors compared to *MAE*. In particular, *RMSE* is widely used for evaluating ML and DL model performance. However, *MSE* is sometimes preferred because it simplifies the mathematical operations compared to *MAE* and *RMSE*. Regarding *R*^2^, it is primarily employed for explaining how well the independent variables in the ML and DL models explain the variability in the dependent variable. To summarize, in the context of regression model performance, while lower values of *MAE*, *MSE*, and *RMSE* indicate better performance of the regression model, a higher value of *R*^2^ is always desirable.

#### 2.2.2. Metrics for Classification Models

In the context of classification models, the evaluation typically relies on metrics such as *accuracy*, the confusion matrix (CM), the area under the curve (AUC) and the receiver operator characteristic (ROC), *precision*, *recall*, *F1 score* (*F1*), and the Matthews correlation coefficient (*MCC*) [41,42].

*Accuracy* is the most familiar metric. Its score is in the range 0 to 1 and represents the number of correct predictions made by an ML or DL model divided by the total number of predictions. This metric should rarely be used in isolation. Particularly in the case of imbalanced datasets, where one class significantly outnumbers another, the accuracy can be misleading. It is expressed as in Equation (8):(8)$$\;Accuracy=\frac{Correct\;Predictions}{Total\;Predictions}$$

The CM is a useful tool that visually represents the model’s prediction. It compares the number of predictions for each class that are correct and those that are incorrect. It involves four variables:True positives (*TPs*): Number of positive observations the model correctly predicted as positive.False positives (*FPs*): Number of negative observations the model incorrectly predicted as positive.True negatives (*TNs*): Number of negative observations the model correctly predicted as negative.False negatives (*FNs*): Number of positive observations the model incorrectly predicted as negative.

An outcome from a CM could look like Figure 2, where a logistic model has been applied to a two-class classification task. The classes are labeled as 0 (positive) and 1 (negative). The true positives (*TPs* = 197) denote that the classifier accurately identified 197 instances as positive. In contrast, the true negatives (*TNs* = 220) indicate the model’s correct classification of 220 instances as negative. The false positives (*FPs* = 43) refer to the 43 instances that the model incorrectly identified as positive when they were actually negative. Lastly, the false negatives (*FNs* = 40) represent the model’s error in labeling 40 positive instances as negative. Together, these values provide a comprehensive overview of the classifier’s performance and areas for potential refinement.

The ROC is a probability curve that plots the true positive rate against the false positive rate for different threshold values. The AUC measures the classifier’s ability to distinguish between classes with a score that ranges between 0 and 1. An example of an AUC/ROC can be observed in Figure 3. The model achieved a score of 0.93, which is equivalent to the area under the orange curve. This high score indicates the model’s good performance in distinguishing between the classes.

*Precision* is a metric that measures how accurately the model identifies the positive class, or in other words, for all predictions of the positive class, how many were correct (see Equation (9)). This metric’s value ranges between 0 and 1, with 1 indicating that all the predicted positive instances for a given class were actually positive and 0 indicating none were, and also this metric allows the false positives of the model to be minimized. The *precision* for class 0 in the example given is 0.83, while for class 1, it is 0.84; these values are relatively high, suggesting that the model’s predictions for both classes are accurate.
(9)$$\;precision=\frac{TP}{TP+FP}$$

On the other hand, *recall* measures the model’s effectiveness at correctly predicting all the positive observations from the dataset (see Equation (10)). *Recall* values also range from 0 to 1, with 1 indicating that the model identified all actual positive instances for a class and 0 indicating it did not identify any. Since *recall* excludes information about the *FP*, it is appropriate when the focus is on minimizing false negatives. Referring again to the given example, the *recall* metric value for class 0 stands at 0.82, while for class 1, it is 0.85, demonstrating a high ability to correctly classify positive instances for both classes.
(10)$$\;recall=\frac{TP}{TP+FN}$$

The *F1 score* combines the precision and *recall* into a single measure that captures both properties, giving a number between 0 and 1 that summarizes the model performance. The higher the value of *F1*, the better the performance of the model. The *F1 score* is calculated as the harmonic mean of the two fractions, as shown in Equation (11). Using again the given example, the *F1 score* metric’s value for class 0 is 0.83, while the *F1* value for class 1 is 0.84, and this signifies a strong balance between *precision* and *recall* for both classes, suggesting a model that is performing well both in terms of avoiding false positives and identifying true positives.
(11)$$\;F1\;score=2\;*\;\frac{precision\;*\;recall}{recision+recall}$$

The *MCC* is another metric for evaluating classification models. This metric, unlike any of the previously mentioned metrics, considers all possible prediction outcomes, taking into account true and false positives as well as true and false negatives. Therefore, it accounts for imbalances in classes. It is in essence a correlation coefficient between the observed and predicted classifications. The *MCC* ranges between −1 and 1, and it is defined as in Equation (12). The *MCC* value obtained for the example given is 0.67, denoting a strong positive correlation between the model’s predictions and the actual classifications, signaling a good-quality model. However, there is still room for improvement as the value has not reached its optimum of 1.
(12)$$\;MCC=\frac{TP\times TN-FP\times FN}{\sqrt{\left(TP+FP\right)\left(TP+FN\right)}\left(TN+FP\right)\left(TN+FN\right)}$$

Our initial methodology, outlined in Section 2, emphasized the importance of diverse evaluation indices such as CM, AUC, *MCC*, *RMSE*, *R*^2^, among others, and a predominant reliance on *accuracy* as the primary performance metric was observed in the referenced studies. *Accuracy* provides a valuable metric but with a limited perspective on a model’s performance, especially in scenarios where data distributions are imbalanced, as occurs with optical spectroscopy datasets. This absence of metrics imposes a limitation on our analysis, particularly as the nuances of model behavior are better captured by a combination of metrics, especially in edge cases and imbalanced datasets.

## 3. DA to Enhance the Dataset

DA can be beneficial for DL problems for a few reasons. First, the model will be trained using a larger and more varied dataset. Second, it helps to reduce the overfitting. DA also increases the efficiency of the training process, since the model will learn from more data in each epoch, which is a single pass through the entire training dataset during the training process. Finally, DA helps to improve the generalizability of the model, as it will be exposed to a greater variety of data points.

Regarding DA implementation, it is often achieved by simply adding noise to the data or by randomly perturbing the existing data points. There are a few more advanced methods for DA that are based on DL, such as using a generative model like a GAN to generate new data points. Both simple and more advanced methods will be reviewed in the next subsections.

### 3.1. DA Based on Non-DL Algorithms

The improvement of the data diversity and quality, ultimately leading to better-trained ML models, can be achieved by duplicating existing instances or introducing new synthetic instances. One way to add more data to an optical spectroscopy dataset consists of determining the standard deviation for each wavelength, selecting the noise scaled according to the standard deviation, and adding Gaussian noise to the spectra at each individual wavelength. In [43], the technique of adding noise to a spectral dataset was explored, where the authors tackled a critical problem for the chemical process industry: monitoring the end group properties in an industrial batch polymerization process with the data collected from an NIR probe [43]. Their aim was to develop robust inferential models that could predict product quality. The authors proposed a partial least squares (PLS) model [44]. The PLS model was trained with a dataset where the samples were enhanced with Gaussian noise, calculated as a percentage of the standard deviation of each wavelength in the original dataset. The results obtained are presented in Table 1, and it is possible to see how the *RMSE* for validation decreases when the number of synthetic data increases. However, the *RMSE* for calibration, after adding 15% scaled noise, begins to rise. This indicates that excessive noise can distort the original samples, and the PLS model struggles to classify them accurately.

Noise can be tailored to develop more robust inferential models. It can be like the noise that is generated by various sources, including variations in the sample being analyzed, or changes in the optical setup used in the experiment, as described in [45]. For example, additive noise simulates the effects of spectra baseline shifts or background differences that may be caused by variations in the sample being analyzed. This can be simulated using Gaussian noise with zero mean, adding this value to all variables in the spectrum. Sample packing, particle size differences, or changes in the optical path length can be simulated by multiplying log-normal noise with a mean of 1, which will ensure that the multiplicative noise is positive.

In [46], the authors developed a DL algorithm, based on a CNN, to predict the drug content in tablets using NIR spectra. The results obtained revealed that the proposed CNN model outperformed a PLS model across all preprocessing combinations, with the best results obtained when DA was applied to the dataset; see Table 2.

The research presented in [47] studied coffee, an economically important commodity frequently adulterated for economic gains, that is negatively affected by food frauds. In this sense, this study evaluated the feasibility of applying convolutional neural networks (CNNs) in comparison to classical chemometric algorithms, such as PLS and interval partial least squares (iPLS) [48] for predicting coffee adulteration. Here, a commercial ‘espresso’ coffee sample was mixed with chicory, barley, and maize. Then, Fourier transform infrared (FT-NIR) [49] spectral data were acquired from the prepared samples, and DA with autoscaling (AS) and/or standard normal variate (SNV) pre-treatment was applied. The spectral acquisition was carried out using a laboratory-based FT-NIR analyzer (Antaris II from Thermo Fisher Scientific, Wisconsin, USA, based on an InGaAs sensor) operating in the spectral range of 1000–2500 nm with a resolution of 4 cm^−1^. The results are shown in Table 3, where it is possible to see how the best model performance is obtained when the DA is applied. For chicory, the CNN model with DA + AS preprocessing has the best performance according to the *RMSEP* (the lower value, *RMSEP* = 0.76) and *F1 score* (the higher value, *F1* = 0.99) metrics. For barley, the best model is obtained with DA + SNC + AS preprocessing with an iPLS model (*RMSEP* = 0.60 and *F1* = 1.00), and for maize the best model is with DA + SNV + AS preprocessing and an iPLS model (*RMSEP* = 0.71 and *F1* = 0.99). The researchers emphasized the advantages of DA in creating better and more robust models, such as adding random offset, multiplication, and slope effects to the raw spectral data, based on the standard deviation of the samples collected.

The research in [50] is focused on the detection of citrus black spot disease and ripeness level in orange fruit using deep neural networks incorporating DA techniques. Citrus black spot disease is a fungal infection that affects citrus fruits, especially oranges, and can cause significant losses in production. The ripeness level of the fruit is also important for ensuring optimal quality and taste at the time of harvest. The researchers in this study worked with a dataset of images from oranges in four categories: unripe, half-ripe, ripe, and infected. The images of the dataset were collected employing a 13 MPx camera (4182 × 2322 pixel size) based on a CMOS BSI sensor, with horizontal and vertical resolutions of 72 dpi, and a standard RGB color space. The dataset was increased using an effective noise-based DA approach (Gaussian, speckle, Poisson, and salt-and-pepper noise). An example of the sample generation using noise-based DA is shown in Figure 4.

The authors used an optimization algorithm to identify the best noise parameters, and several pre-trained models (GoogleNet, ResNet18, ResNet50, ShuffleNet, MobileNetv2, and DenseNet201 [51]), and the authors suggest that this technique can be easily applied to detect other diseases. The results of the study can be seen in Table 4.

The identification of vegetable oil species in oil admixtures for food analysis was explored in [52] with the aim of enhancing the robustness of the model. Here, three datasets with FT-NIR samples were created with different mixtures of vegetable oils, and the following four DA techniques were applied to improve the number of samples:Blending spectra: combines samples with variations to create artificial admixtures.Spectral intensifier: modifies the intensity of a spectrum to control baseline variations.Shifting along the *x*-axis: applies random shifts to data points in spectra to mimic instrumental variations.Adding noise: increases the variability of a class by including slightly noisy spectra based on the original spectra.

Previous DA techniques can be applied independently or in combination, with adjustable input parameters, to create the desired variability in the dataset. In Figure 5, it is possible to see how DA techniques modify the spectral data.

The results of the study are shown in Table 5, showing an improvement in the classification *accuracy*. The results show how the classification rate for the testing dataset without DA is 63%. However, when the technique of spectral intensifier (between 1% and 33%) is applied in combination with shifting along the *x*-axis (shift equal to 0.6) and Gaussian noise (35 dB), there is an improvement in the *accuracy*, which rises to 88%.

A different DA technique is the synthetic minority oversampling technique (SMOTE), which is presented in the paper titled “SMOTE: Synthetic Minority Over-sampling Technique” [53]. SMOTE is a simple approach to improve the results of traditional ML and algorithms. It synthesizes new data based on the existing data, oversampling the minority class with the aim of improving the *accuracy* for classification tasks and reducing the losses in regression models. SMOTE works by selecting samples that are close to each other in the feature space, placing a line between the samples in the feature space and adding a new sample at a point along that line. The process begins with the selection of a random sample from the minority class. Then, k of the nearest neighbors for that sample are found and one of them is randomly selected. Finally, a new synthetic sample is created at a randomly selected point between the two samples selected from the feature space. Figure 6 represents the working principle of SMOTE, where the red triangles represent the minority class, and black arrows are the lines in the feature space where the synthetic data will be added.

The research in [55] demonstrates how the utilization of SMOTE with VIS–NIR diffuse reflectance spectroscopy helps to predict soil properties in situ. The samples in this work were obtained using an AgriSpec portable spectrophotometer equipped with a contact probe (from Analytical Spectral Devices, Boulder, CO, USA) based on an InGaAs sensor that permits the samples to be scanned in the spectral range of 350–2500 nm with 1 nm intervals. The researchers propose the utilization of SMOTE for the generation of synthetic samples to calibrate a PLS model for organic carbon soil prediction. The proposed model improves the prediction *accuracy*, compared to the model calibrated using only air-dried samples, as shown in Table 6. It is expected that *R*^2^ will be bounded between zero and one when a linear regression model is fit and evaluated on the same data it is fitted to. Otherwise, *R*^2^ can lead to negative values. The researchers suggest that the negative *R*^2^ values might be attributed to the SMOTE method employed, where the manner in which the synthetic samples were generated and the number of neighbors employed could be significant factors contributing to the negative *R*^2^ value.

Another example of the utilization of DA in order to balance datasets is presented in [54]. Here, different diesel oil brands are rapidly and accurately identified, which is crucial for the proper operation of diesel engines. The researchers in this work use NIR spectra that can be classified into five different categories, but not all of them have the same number of samples, this is known as an unbalanced dataset, and has a negative impact on the performance of AI and DL algorithms [56]. The authors exploit the SMOTE technique to balance the dataset, and, as is presented in Table 7, the model performs well, accurately identifying diesel oil brands even with an unbalanced dataset.

For the sake of simplicity, Table 8 summarizes the main aspects of the works explained above, along with information on the application domain, the spectral range of the analyzed samples, and the specific DA technique employed in each case.

### 3.2. Generative Adversarial Networks (GANs)

GANs are a type of unsupervised machine learning algorithm aimed at generating new data samples (termed fake data) that are close to the data from the dataset (termed real data). This algorithm consists of two neural networks, named the generator and the discriminator, that compete in a zero-sum game. The generator crafts fake data samples that mimic the real data that could be part of the real data dataset. In contrast, the task of the discriminator is to determine if the sample is genuine (a sample from real data) or if it is a fake one (created by the generator). This adversarial game continues until the data produced by the generator is indistinguishable from the real data [57].

GANs’ two main components (generator and discriminator) are schematically represented in Figure 7. Here, the generator receives random noise as input and generates synthetic data samples. The discriminator takes in both real data samples and the synthetic samples created by the generator, with the main task of classifying each input as real data or produced by the generator. The output of the discriminator consists of two blocks, one for true (real data) and one for false (generated data). The error from these classifications is then backpropagated through both the generator and discriminator during the training process, allowing them to continuously improve their performance in a competitive manner [58].

GANs [59] have been used to generate new images [60], videos [61], text (known as natural language processing) [62], voices [63], and they have also been used to generate fake faces [64]. Recent advances from the original GAN presentation [58] have led to the development of new architectures such as convolutional GAN (C-GAN) and deep convolutional GAN (DC-GAN) [65,66,67], semi-supervised GAN (SGAN) [68], CycleGAN [69], and bidirectional GAN (BI-GAN) [70], among others. These architectures have been demonstrated to be able to generate high-quality images or text and improve the stability of the training process.

An example of the application of GANs in the agrifood industry is shown in [71]. In this study, the authors proposed an innovative, rapidly, and non-destructive methodology that permits the oil content of individual maize kernels to be predicted from hyperspectral images of two varieties of maize kernels (named Zhengdan958 and Nongda616), which is a crucial task in this field. Image acquisition was achieved using an N17E spectral camera from Spectral Imaging Ltd., based on an InGaAs sensor, that allows operation in the NIE or SWIR spectral range, from 900 to 1700 nm. Since obtaining a large volume of counter reference values for maize kernels is time consuming and very expensive, the authors explored a combination of hyperspectral images and DC-GAN in order to expand the spectral data and oil content data. DC-GAN was specifically employed to generate synthetic data to obtain a large training dataset and improve the generalization ability of the model. Figure 8 shows how DC-GAN can synthetize new data as the training process is running.

The performance of two different regression models (PLSR and SVR) was compared before and after DA. The results are shown in Table 9, which confirm that DA techniques enhanced the performance of both regression models, providing a solution for the lack of data.

A different study utilized a combination of NIR spectroscopy and GANs to identify cumin and fennel samples from different regions, which is very important for the quality control and food process as well as to avoid market fraud [72]. The research deals with the precise identification of cumin (*Cuminum cyminum*) and fennel (*Foeniculum vulgare*), employing ML and DL algorithms combined with NIR spectroscopy. Here, cumin and fennel samples collected from different regions were measured using a VERTEX 70 FT-IR spectrometer, with a CO_2_ compensation parameter, in the spectral range of 4000–11,000 cm^−1^ with a resolution of 8 cm^−1^ and 32 scans. Spectral data dimensionality reduction was performed using different techniques, such as principal component analysis (PCA) and rubberband baseline correction. A reduced dataset was used to develop classification models, such as quadratic discriminate analysis based on PCA (PCA-QDA), ANN based on PCA (PCA-MLP), and CNN based on the spectral data. A schematic representation of this process is presented in Figure 9.

The *accuracy* results obtained from the study are shown in Table 10, where the GAN model outperforms the PCA-QDA, PCA-MLP, and CNN, reaching a classification *accuracy* of 100%.

Another study proposed an early detection method for tomato spotted wilt virus (TSWV), which is one of the most important diseases affecting tomatoes, allowing for timely intervention and disease management and contributing to enhanced crop health and productivity [73]. Here, the researchers aimed to develop an early detection method of TSWV in tomato leaves by means of the combination of hyperspectral imaging and a modification of the GAN architecture to remove outliers from the input images [74]. The proposed method, called outlier removal auxiliary classifier generative adversarial nets (OR-AC-GAN), incorporated tasks such as plant segmentation, spectrum classification, and image classification by focusing on image pixels. The results obtained in this work are shown in Table 11, and this model has the potential to be applied to other plant disease detection applications. The validation of the proposed model was carried out with a wide-spread plant disease TSWV, resulting in an *FP* rate as low as 1.47%, and a sensitivity and specificity of 92.59% and 100%.

The study in [75] is focused on the rapid identification of marine pathogens in marine ecology, with challenges associated with the high cost and challenging process of sample collection because of the nature of the experimental environment. The researchers propose a novel method in this work, combining a Raman spectroscopy dataset with GANs to classify three different marine pathogens: *Staphylococcus hominis*, *Vibrio alginolyticus*, and *Bacinillus licheniformis*. A schematic representation of the experimental workflow is presented in Figure 10.

Also, the study attempted to identify potential distinctive regions in the Raman spectra, resulting in a promising tool for pathogen identification in marine ecology using Raman spectroscopy. The results given in the research are presented in a CM with 100% prediction *accuracy* for all the pathogens under study.

Raman spectroscopy combined with GANs and a multiclass SVM is also explored in [76] for the rapid detection of the foodborne pathogenic bacteria *Salmonella typhimurium*, *Vibrio parahaemolyticus*, and *E. coli*, associated with food safety testing. A schematic representation of the experimental workflow is presented in Figure 11.

Here, the utilization of GANs can solve the problem of requiring vast sample quantities for the training sets, while enhancing the classification *accuracy* of the model. In particular, this approach permitted the parameters of the SVM to be optimized and obtained improved classification results. The results are presented in Table 12, which gives a view of the power of Raman spectroscopy combined with AI in order to develop accurate, efficient, and effective food safety models.

Previous methodologies combining optical measurement with GANs could be easily applied to other agricultural products, contributing to quality control, authentication processes, waste reduction, food safety, and production increase in the agrifood industry.

#### Semi-Supervised Learning—SGANs for DA and Classification

SGANs are a type of GAN that can learn from both labeled and unlabeled data [77], allowing them to capture a broader understanding of the data distribution and generate more realistic data [69]. SGANs differ from the original GANs in the way they use labeled data. While GANs are unsupervised algorithms using unlabeled data, SGANs utilize labels within the training process. Hence, they are termed as semi-supervised [78]. The architecture of SGANs, like typical GANs, involves a generator and a discriminator competing in a zero-sum game, but the discriminator in SGANs is divided into a supervised (sup.) and unsupervised (unsup.) neural network (see Figure 12). The supervised discriminator is a multiclass classifier used to categorize the “labeled real data” according to its respective labels, while the unsupervised discriminator works as a binary classifier, trained to differentiate between the “unlabeled real data” and the data generated by the generator [79].

The SGAN training process begins with “labeled real data”, allowing the discriminator to learn the underlying data structure of the samples, similar to the approach of a multiclass classification algorithm. The predictions of the discriminator are compared with the corresponding labels (labels—gray box), and the weights and bias of the discriminator are updated through the backpropagation method. Then, “unlabeled real data” is utilized to fine-tune the learned features and enhance the quality of the generated data [80]. Given the lack of labels for the latter, the discriminator is trained as a binary classification, distinguishing between true (unlabeled real data) and false (fake data).

The losses observed during the training of the discriminator are then used to guide the training of the generator. The objective is to minimize these losses, so the weights and bias are updated accordingly using the backpropagation method. After each training iteration, the generator model will be able to produce more realistic (fake) data. In order to achieve an effective SGAN training process, this cycle of training the discriminator with both sets of real data, and subsequently training the generator, must be repeated continuously [69].

SGANs can be used in many fields, including medical imaging, where they are used to identify anomalies for disease progression and treatment monitoring [81] and for medical image segmentation [82,83], or in industrial applications, where they can be used to solve the problem of imbalanced data for applications where data collection can be costly and difficult [84].

SGANs have demonstrated their versatility across various disciplines for classification [85] or regression [86] tasks. Due to their ability to learn from both labeled und unlabeled data, SGANs work well with images, as they can extract the natural structure of the images to learn patterns and features [29]. These properties make them a suitable architecture to be used with hyperspectral data, where the inherent structure and features of the data can be utilized to improve the quality of the generated samples. The application of SGANs in the agrifood domain was presented in [87] for precision agriculture monitoring of farmlands, emphasizing the importance of intelligent methods in identification to protect the fields from weed infestations. Here, unmanned aerial vehicles (UAVs), equipped with four-band Sequoia multispectral 16 Mpx CMOS cameras, were used for image acquisition. The researchers used the WeedNet dataset, a GNU General Public License dataset with multispectral images of crop/weed fields [88], to train and evaluate their proposed method based on an SGAN architecture with a DC generator with labeled and unlabeled data, to differentiate between crops and weeds. Sample images used to train the SGAN model are shown in Figure 13.

The obtained results are shown in Table 13, demonstrating the potential of DC-GAN to generate hyperspectral images (HSIs) to enhance precision agriculture. It shows how the classification performance of the model improves as the number of labeled images increases. This might happen because the discriminator receives more direct guidance on distinguishing between different classes, and if the labeled dataset is not diverse enough there is a risk that the discriminator might overfit to this labeled data. If the discriminator improves with more labeled data, the generator might also improve, and the training process could be quicker, because the models might converge faster with more labeled data. Even so, by decreasing the amount of unlabeled data, it is also decreasing the key advantage of an SGAN, its ability to leverage a large amount of unlabeled data.

Also focused on precision agriculture, the research in [89] explored how to identify weed crops in a field for the proper usage of pesticides and chemicals. The proposed image acquisition system consisted of a simple quadcopter equipped with a red, green, blue (RGB) camera operating in two different croplands (pea and strawberry). Here, the researchers used a pre-trained neural network to make classifications, and the best results were obtained with ResNet50 [51], as shown in Table 14. Other models were tested with similar results, but higher values were obtained with ResNet50. The choice between models will depend on the specific requirements of the computational resources available more than the performance.

Another example of the use of SGANs with HSIs for DA in order to enhance the performance of the model is presented in [90]. In this work, the researchers evaluate the performance of their proposed model with the Indian Pines dataset, a dataset that has been made with an airborne visible/infrared imaging spectrometer (AVIRIS), an optical sensor that captures data in the range from 400 to 2500 nm. In the study, the authors propose a novel semi-supervised algorithm for HSI data classification, based on a 1D GAN (named by the authors as HSGAN), that facilitates the automatic extraction of spectral features for classification. The results, presented in Table 15, show that the proposed model achieves promising results, even with limited data, setting a new working method with HSIs. To evaluate the model, the researchers look at two metrics, the overall *accuracy*, which is the total number of correct predictions divided by the total number of samples, and the average *accuracy*, the average of the accuracies for each individual class in a multiclass classification task. As both the *overall accuracy* and the *average accuracy* approach 100%, the performance of the model improves.

The research in [91] also focuses on the classification of high-dimensional HSIs with a limited number of training samples. The method proposed here consists of two main ideas. First, a three-dimensional bilateral filter (3DBF), a powerful tool for image applications, is prepared for a feature extraction of the HSI data, allowing the high dimensionality of the HSI images to be reduced. Second, an SGAN is trained with the filtered HSIs, where the GAN model is compared with HSIs (named spec) versus the 3DBF method for three different datasets, the Indian Pines dataset, University of Pavia dataset, and Salinas dataset. The Indian Pines and Salinas datasets are both made with the same sensor (AVIRIS), but capturing over a different area. The optical sensor used to capture hyperspectral data from the University of Pavia employs a reflective optical system imaging spectrometer (ROSIS), that measures in the range from 430 to 860 nm. Figure 14, Figure 15 and Figure 16 show sample images from the mentioned datasets. On the left side of the figures there is a fake color composite of the data captured by the UAVs, and on the right side there is ground truth data with the different classes of crops and fields.

The effectiveness of the proposed method is confirmed by the experimental results shown in Table 16, which reveals the effectiveness of the method with a limited number of labeled samples.

Table 17 summarizes the main aspects of the works explained above in order to facilitate the usage of this review by the reader, along with information on the application domain, the spectral range of the analyzed samples, and the specific DA technique employed in each case.

### 3.3. Comparative Analysis of DA Techniques: Merits and Disadvantages

In the domain of agrifood analytics, the use of electromagnetic spectroscopic data in the UV, VIS, and NIR regions is becoming increasingly common. However, one major challenge is to build robust ML models. DA is as a viable solution to this problem, enhancing models’ performance by artificially increasing the size of datasets. This review outlines various DA methods based on non-DL algorithms, such as noise addition or using the SMOTE and DL algorithms, particularly focusing on GANs and SGANs. Table 18 briefly outlines the primary advantages and disadvantages of these methods without delving into details, especially when applied to spectroscopic data in agrifood analytics.

To summarize:Non-DL-based methods: The explained methods encompass the addition of various types of noise to the existing data: shifting along the wavelength axis, modifying the spectral intensity of the samples, or applying SMOTE, among others. These methods are useful and simple to apply but may not achieve the precise discernment of spectral features if not applied correctly, resulting in a lack of real semantic variability. This means that the diversity in the generated data can be non-meaningful or irrelevant to the task.GAN and SGAN methods: Their main strength is their ability to learn complex data distributions to generate more realistic data with real semantic variability. In contrast, these methods require considerable expertise in tuning and training, as well as the availability of computational resources.

Each method comes with its owns set of tradeoffs, and the choice between them should be dictated by the specific requirements of the spectroscopic analysis in the agrifood sector, the available computational resources, and the expertise of the researchers involved in the project.

## 4. Conclusions

This work has presented a comprehensive review of DA, from the simplest to the more complex methods, in order to enhance the understanding of how DA can be used as a powerful tool for future work. The study has focused on the use of DA in applications working with optical spectroscopic data, applied to the agrifood industry. The aim of the DA techniques reviewed in this work is to develop more robust, and better performance models for predicting or classifying the properties or qualities of samples as well as distinguishing diseases or pathogens, with a focus on the agrifood industry.

Two main DA techniques have been studied in this work. The first DA technique is based on non-DL algorithms, which require a detailed analysis of the dataset (see Section 3.1), and show great results in the improvement of models. The second DA technique is based on using DL algorithms to enhance the datasets, specifically those algorithms based on GANs and SGANs, which have proven to be a great technique for enhancing prediction models (see Section 2.2) based on HSIs.

In particular, the main advantage of the GANs is their ability to produce fake data that mimics the original dataset, and, with enough time, fake data cannot be distinguished from data obtained from real samples. The advantage of the SGAN architecture is that it allows a generator model to be trained to mimic samples, and a classification model to predict between different classes, at the same time.

DA techniques can help to improve the results obtained with optical spectroscopy datasets related to the agrifood industry when there is no possibility to obtain a complete or balanced dataset. Thus, it will be our task to take advantage of the DA techniques mentioned above to develop a proper dataset for our study. In order to provide a generic methodology for the reader to take advantage of DA techniques, the following points can be followed:Take a set of samples and perform the optical spectroscopy measurements in the range where the samples present the most relevant information for the target.Apply non-DL-based techniques to enhance the dataset, adding new synthetic data to the dataset. Develop a model and check the performance with these new synthetic data.Apply DL-based techniques (GANs or SGANs) to generate new fake data and evaluate the final model in order to check if there was an improvement in its performance.Release the model and apply it to the agrifood industry particular application case to evaluate the model in situ.

Finally, it is important to remark that DA techniques are presented here as a highly valuable tool for the generation of additional optical spectroscopy data in the agrifood industry. Since datasets in this field are sometimes unavailable, and the process to collect a sufficient number of samples to develop robust prediction models can be costly, time consuming, and inefficient, DA techniques can help to alleviate these problems.

## Figures and Tables

**Figure 1 sensors-23-08562-f001:**
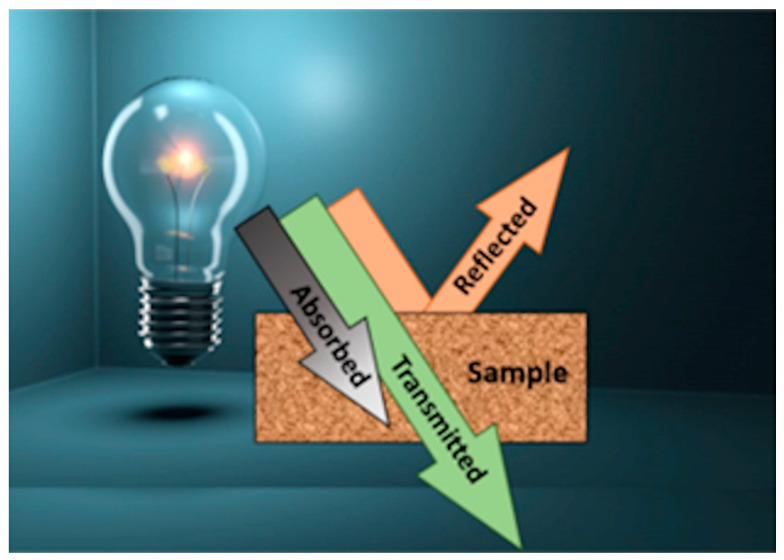
Light interacting with a sample.

**Figure 2 sensors-23-08562-f002:**
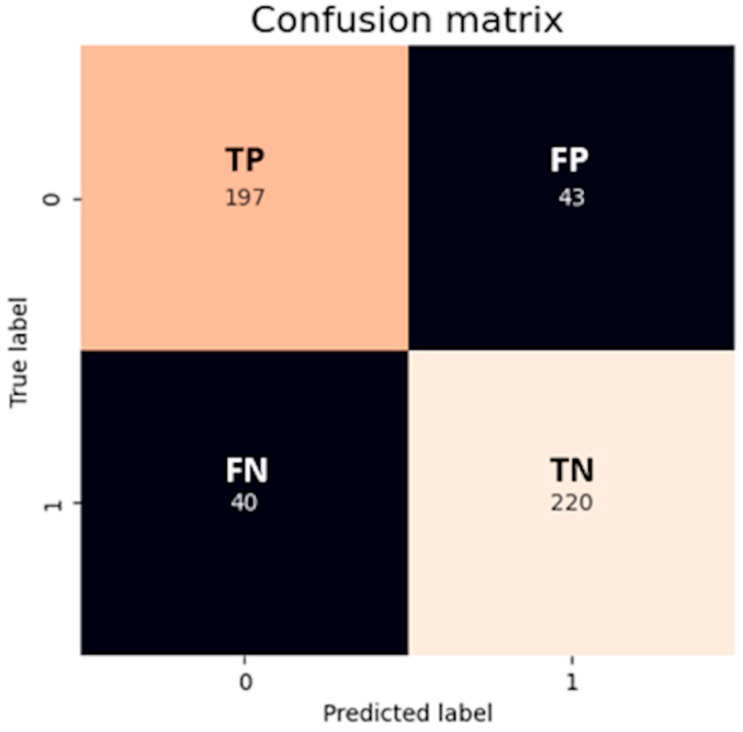
CM for a binary classifier, providing an insight into the classifier’s performance, giving detailed information about its predictions compared to the actual outcomes. Image by author.

**Figure 3 sensors-23-08562-f003:**
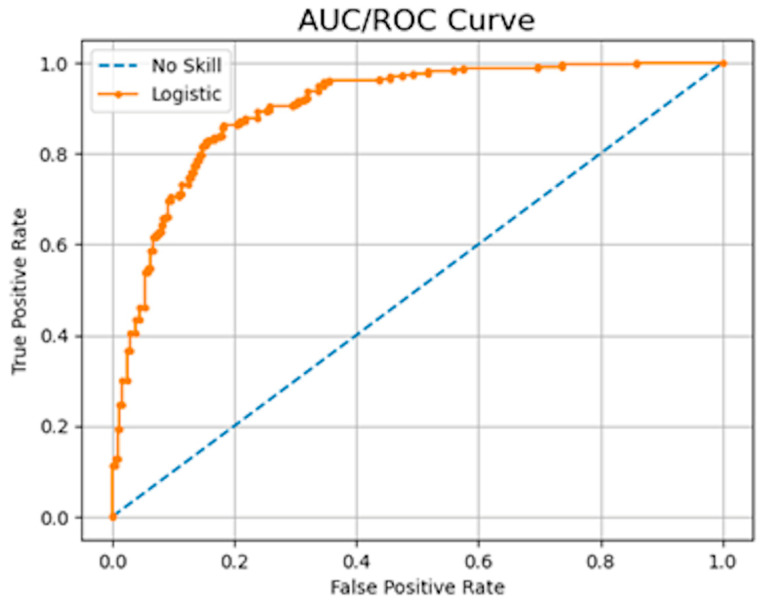
AUC/ROC curve example, showing the model’s ability to discriminate between the positive and negative classes. The AUC indicates the likelihood that the model will rank a random positive sample higher than a random negative sample. An AUC value of 1.0 signifies perfect discrimination, while a value of 0.5 suggests that the model has no discriminative capacity. Image by author.

**Figure 4 sensors-23-08562-f004:**
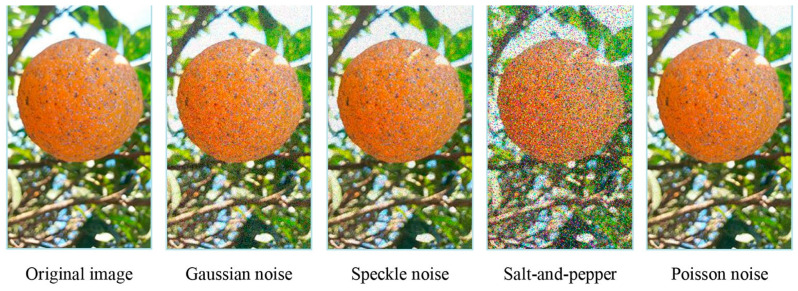
Example of how the application of different types of noise to fresh samples can modify an image of an orange. Reprinted from [50], with permission from Elsevier.

**Figure 5 sensors-23-08562-f005:**
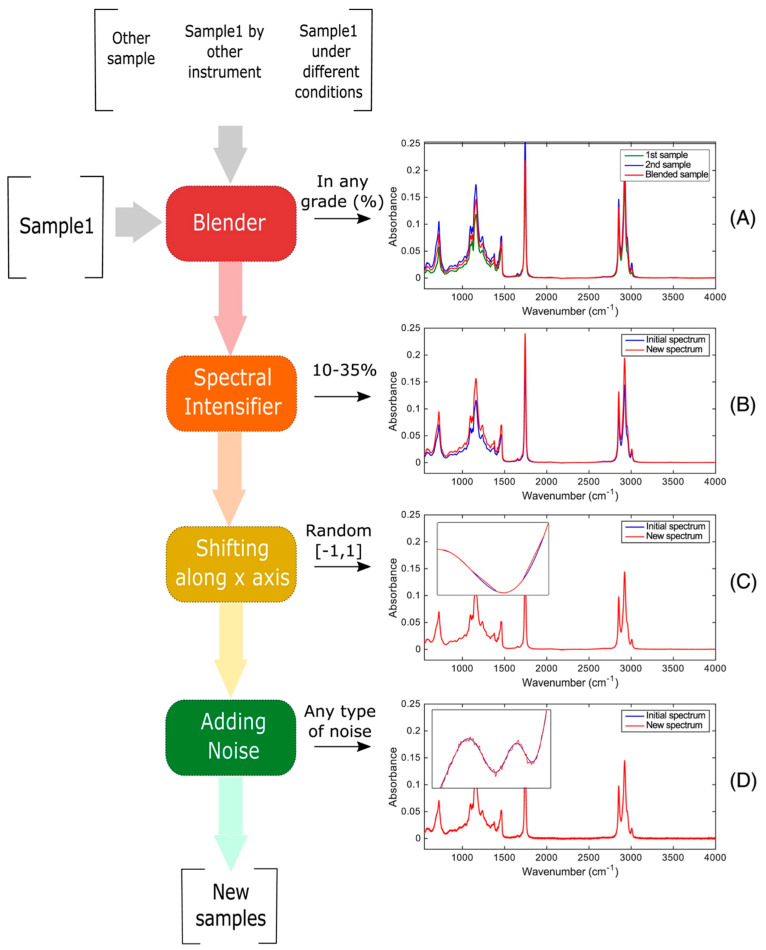
Example of how the DA techniques modify the spectral data. Image from [52] under CC BY Creative Commons License.

**Figure 6 sensors-23-08562-f006:**
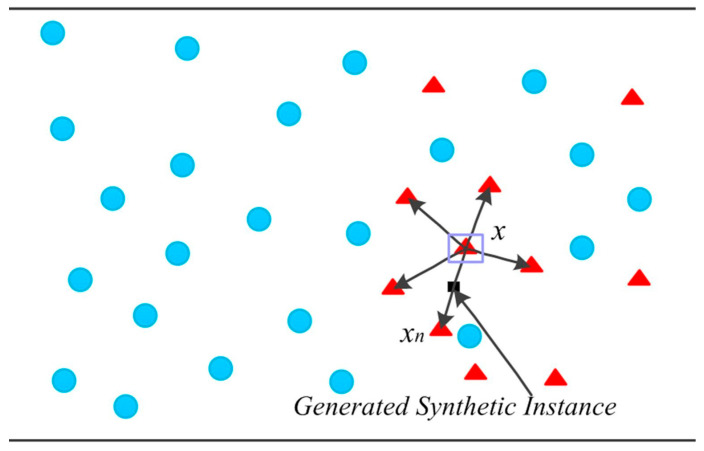
The principle of SMOTE. Image from [54], with permission from Elsevier.

**Figure 7 sensors-23-08562-f007:**
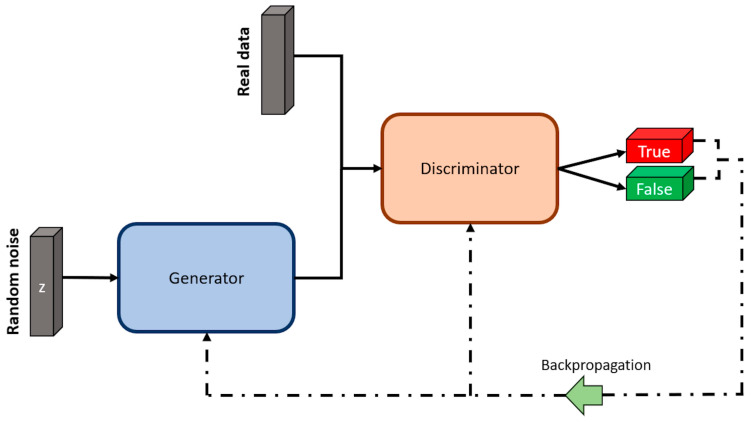
Architecture of a GAN. Image by author.

**Figure 8 sensors-23-08562-f008:**
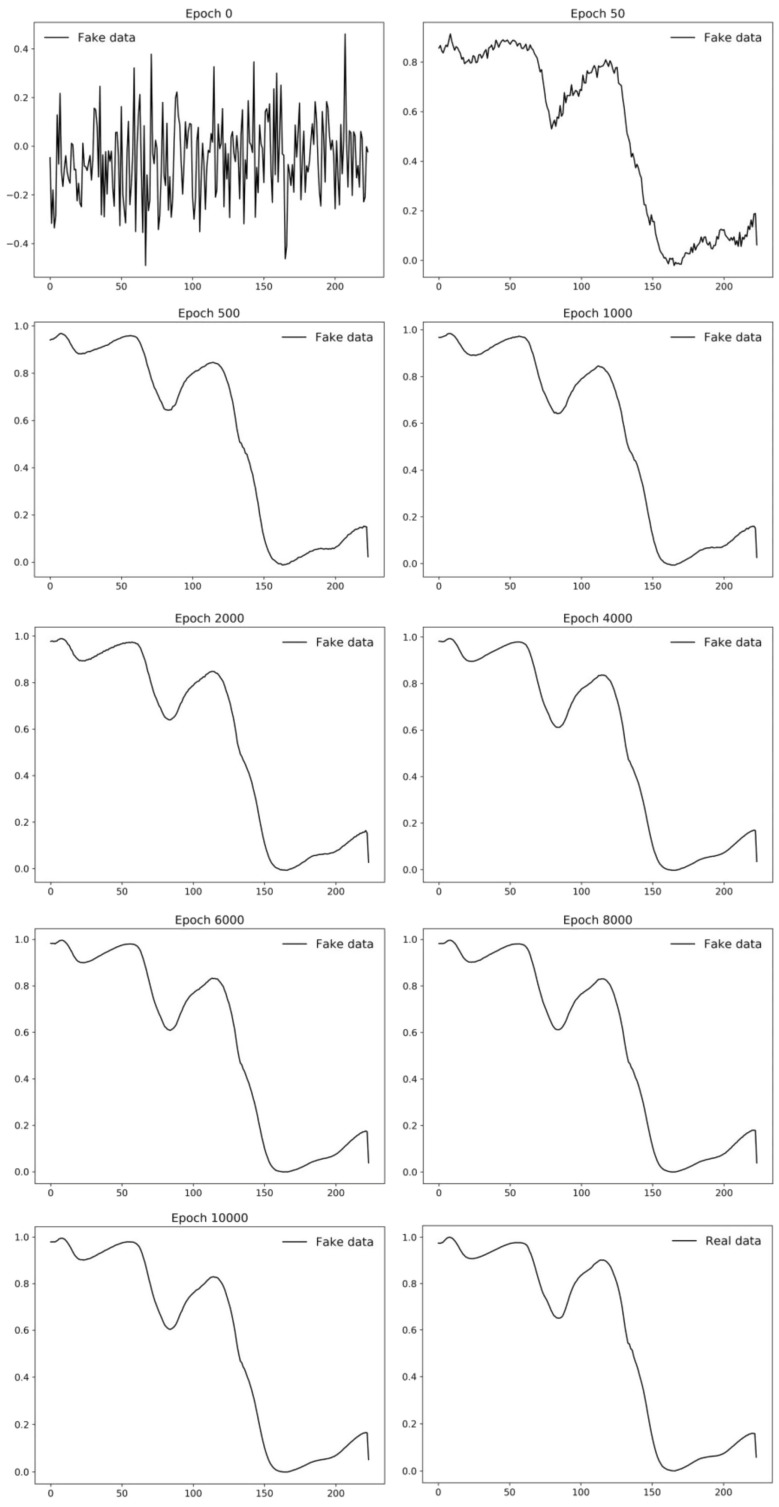
Fake data from GAN at different epochs. Image from [71], with permission from Elsevier.

**Figure 9 sensors-23-08562-f009:**
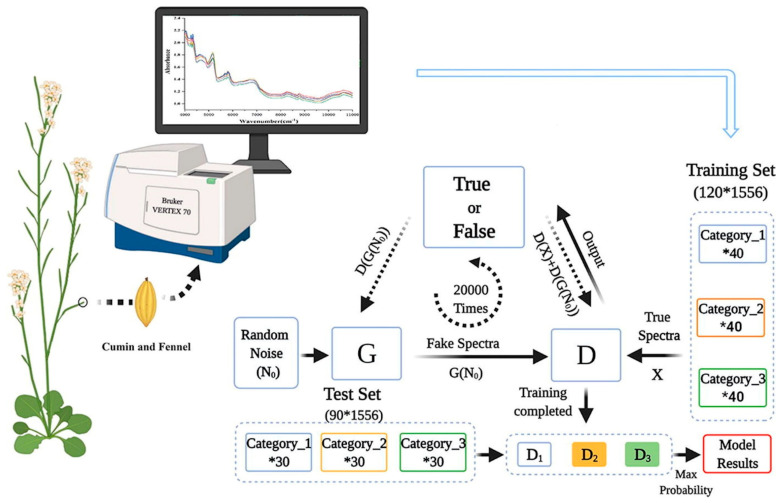
Setup for the precise identification of cumin and fennel. Image from [72], with permission from Elsevier.

**Figure 10 sensors-23-08562-f010:**
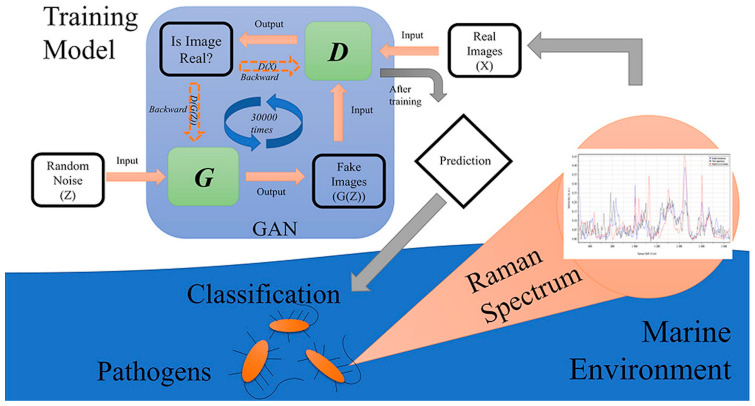
Schematic setup for identification of pathogens in marine ecology. Image from [75], with permission from Elsevier.

**Figure 11 sensors-23-08562-f011:**
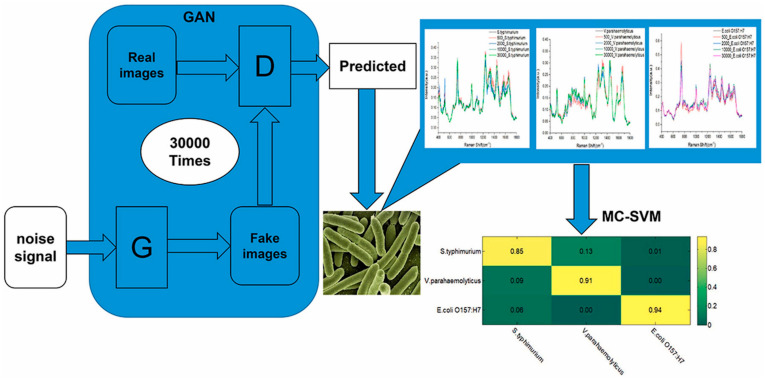
Schematic setup for rapid detection of foodborne pathogenic bacteria. Image from [76], with permission from Elsevier.

**Figure 12 sensors-23-08562-f012:**
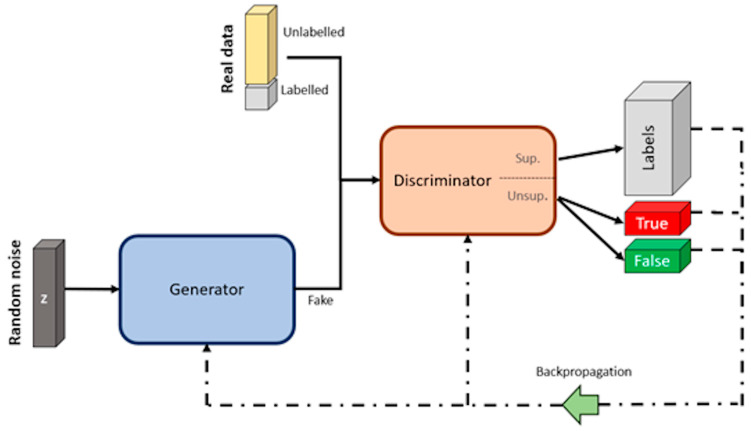
Architecture of SGANs. Image by author.

**Figure 13 sensors-23-08562-f013:**
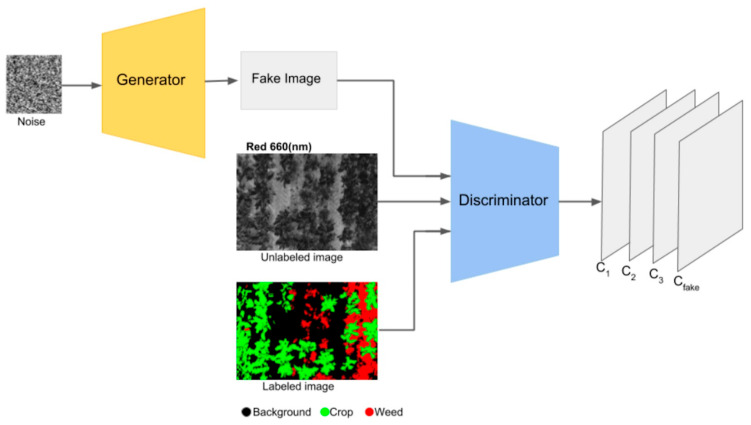
An example from the WeedNet dataset. Image from [87], under CC BY Creative Commons License.

**Figure 14 sensors-23-08562-f014:**
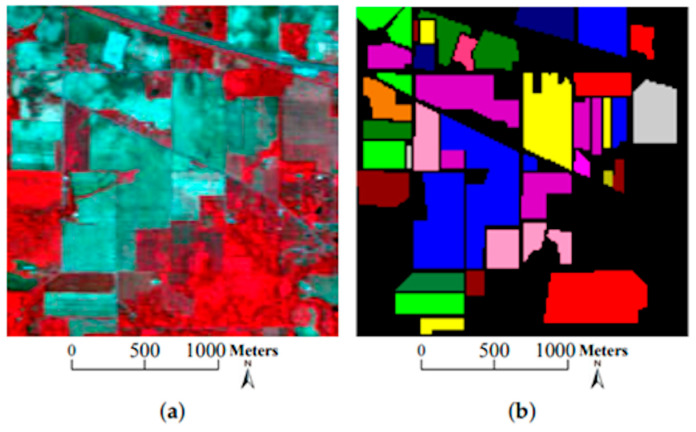
Indian Pines dataset. (**a**) Fake color composite, and (**b**) ground truth data. Image from [91], under CC BY Creative Commons License.

**Figure 15 sensors-23-08562-f015:**
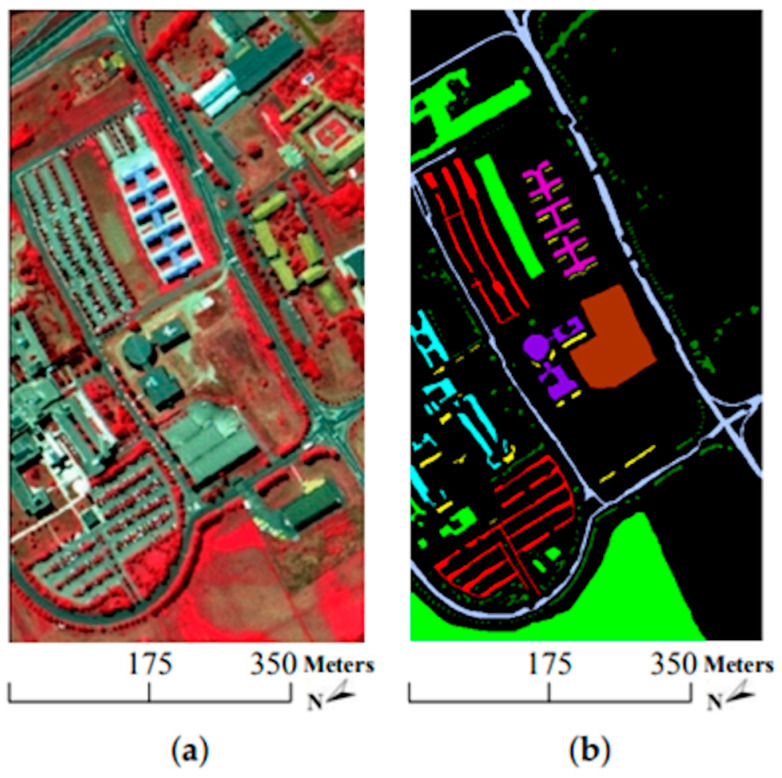
University of Pavia dataset. (**a**) Fake color composite, and (**b**) ground truth data. Image from [91], under CC BY Creative Commons License.

**Figure 16 sensors-23-08562-f016:**
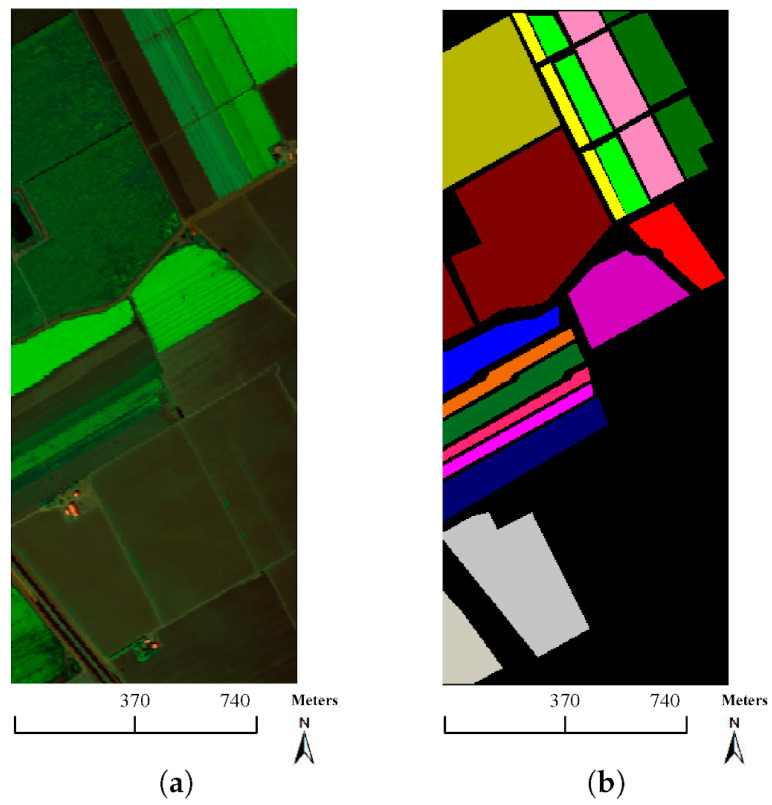
Salinas dataset. (**a**) Fake color composite, and (**b**) ground truth data. Image from [91], under CC BY Creative Commons License.

**Table 1 sensors-23-08562-t001:** *RMSE* for different percentage of noise augmentation.

Noise Augmentation (%)	*RMSE*	
	Calibration	Validation
0	10.96	15.04
5	6.52	17.45
10	5.92	16.95
15	10.10	14.61
20	7.60	14.40
25	13.81	6.52
30	15.76	5.68

**Table 2 sensors-23-08562-t002:** Results predicting drug content in tablets.

		PLS		CNN	
		*R* ^2^	*RMSE*	*R* ^2^	*RMSE*
Without DA	Train	0.97	2.97	0.97	3.02
	Test	0.94	4.43	0.97	4.01
With DA	Train	0.97	3.20	0.98	1.10
	Test	0.95	4.23	0.97	2.44

**Table 3 sensors-23-08562-t003:** Overview of the results for different models and pre-treatments.

Sample Type	Preprocessing	Model	*RMSEP* (%)	*F1 Score*
Chicory (without DA)	AS + iPLS	-	1.06	0.99
Chicory (with DA)	DA + AS	CNN	0.76	0.99
Chicory (with DA)	DA	CNN	1.06	0.98
Barley (without DA)	iPLS + SNV + AS	-	1.06	0.99
Barley (with DA)	DA + AS	CNN	0.80	0.99
Barley (with DA)	DA + SNV + AS	PLS	0.75	0.99
Barley (with DA)	DA + SNV + AS	iPLS	0.60	1.00
Barley (with DA)	DA + SNV + AS	CNN	0.76	1.00
Maize (without DA)	SNV + AS	iPLS	0.73	0.99
Maize (with DA)	DA + AS	CNN	0.82	0.99
Maize (with DA)	DA + SNV + AS	iPLS	0.71	0.99
Maize (with DA)	DA + SNV + AS	PLS	0.83	0.99
Maize (with DA)	DA + SNV + AS	CNN	0.98	0.99

**Table 4 sensors-23-08562-t004:** Best performance of every pre-trained model for citrus black spot disease.

Class	Methodology	*Sensitivity* (%)	*Precision* (%)	*Recall* (%)	*F1 Score*
Orange Infected with Black Spot Disease	ResNet50 with DA	100	100	100	100
Orange (Unripe)	ResNet50 with DA	100	98	100	99
Orange (Half-Ripe)	ResNet50 with DA	98.5	100	98.5	99.3
Orange (Ripe)	ResNet50 with DA	100	100	100	100

**Table 5 sensors-23-08562-t005:** Increase in *accuracy* metrics for the classification of vegetable oils.

Model	Without DA	With DA
PLS	63%	88%

**Table 6 sensors-23-08562-t006:** Improvement on the prediction of soil properties after the calibration with synthetic data.

Model Calibration	*RMSEP*	*R* ^2^
Air-dried spectra only	6.18	−0.53
Air-dried spectra with synthetic field spectra	2.12	0.82

**Table 7 sensors-23-08562-t007:** Results presented for identification of diesel oil brands.

Model	*Accuracy* (%)	AUC/ROC
Tree-XGBoost	80.67	0.78
Tree-SMOTE-XGBoost	94.96	0.97

**Table 8 sensors-23-08562-t008:** Overview of research works using traditional DA techniques.

Application Domain	Dataset	DA Technique	Ref.
F: Adulteration of commercial ‘espresso’ coffee with chicory, barley, and maize.	NIR (1000–2500 nm)	Adding random offset, multiplication, and slope effects.	[47]
F: Identification of vegetable oil species in oil admixtures.	Far IR (2500–18,000 nm)	Noise adding, weighted sum of samples, modify the intensity of the spectra and shift along wavelength axis.	[52]
A: Citrus black spot disease and ripeness level detection in orange fruit.	VIS (440–650 nm)	Noise adding (Gaussian, salt-and-pepper, speckle, and Poisson).	[50]
A: Predicting soil properties in situ with diffuse reflectance spectroscopy	VIS–NIR (350–2500 nm)	Use of SMOTE with different hyperparameters.	[55]
A: Identification of diesel oil brands to ensure the normal operation of diesel engines.	NIR (750–1550 nm)	Use of SMOTE to balance the dataset.	[54]
C: Analysis of drug content in tablets from pharmaceutical industries.	NIR (600–1800 nm)	Adding random variations in offset, multiplication, and slope.	[46]
I: Analysis of an industrial batch polymerization process to monitor end group properties.	NIR (1000–2100 nm)	Noise adding, weighted sum of samples, linear drift adding, smoothing techniques and baseline shifts.	[43]

F: food adulteration; A: agriculture; C: chemical analysis; I: industrial process.

**Table 9 sensors-23-08562-t009:** Results comparing the improvement when using different DA techniques for oil content prediction in two varieties of maize kernel (Zhengdan958 and Nongda616).

	Zhengdan958		Nongda616	
	PLSR	SVR	PLSR	SVR
*R*^2^ Increase (%)	3.47	1.69	3.50	5.34
*RMSE* Decrease (%)	12.78	6.77	12.03	15.29

**Table 10 sensors-23-08562-t010:** Results comparing the model performance for cumin and fennel classification.

	PCA-QDA	PCA-MLP	CNN	GAN
Cumin *Accuracy* (%)	100.0	97.8	96.7	100.0
Fennel *Accuracy* (%)	97.8	96.7	92.2	100.0

**Table 11 sensors-23-08562-t011:** Results for the early detection of TSWV.

	OR-AC-GAN
Plant-Level Classification Sensitivity (%)	92.59
Plant-Level Classification Specificity (%)	100.00
Average Classification *Accuracy* (%)	96.25

**Table 12 sensors-23-08562-t012:** Results for *accuracy* predictions for foodborne pathogenic bacteria.

Bacteria	*Accuracy* (%)
*Salmonella typhimurium*	85.0
*Vibrio parahaemolyticus*	91.2
*E. coli*	94.0
Overall *Accuracy*	90.0

**Table 13 sensors-23-08562-t013:** Results obtained for precision agriculture with UAVs and HSIs.

Labeled Data (%)	50		40		30	
	Crop	Weed	Crop	Weed	Crop	Weed
*F1 scores*	0.857	0.865	0.837	0.834	0.823	0.815

**Table 14 sensors-23-08562-t014:** Results obtained for RGB images obtained with UAVs.

Labeled Data (%)	Model	20%	40%	60%	80%
Cropland Pea (%)	ResNet50	87.67	89.69	92.83	95.98
Cropland Strawberry (%)		88.03	89.91	93.89	96.82
Cropland Pea (%)	ResNet18	87.11	89.42	92.47	95.63
Cropland Strawberry (%)		87.51	89.77	93.47	96.53
Cropland Pea (%)	VGG-16	86.24	89.13	92.23	95.01
Cropland Strawberry (%)		87.13	89.37	93.08	96.21

**Table 15 sensors-23-08562-t015:** Results obtained for an HSGAN.

		PCA-KNN	CNN	HSGAN
5% Supervised Samples	*Overall Accuracy* (%)	71.50	72.00	74.92
	*Average Accuracy* (%)	64.91	66.93	70.97
10% Supervised Samples	*Overall Accuracy* (%)	75.93	78.44	83.53
	*Average Accuracy* (%)	70.76	75.60	79.27

**Table 16 sensors-23-08562-t016:** Results obtained for the classification of HSIs for remote sensing.

	Indian Pines		University of Pavia		Salinas	
	Spec-GAN	3DBF-GAN	Spec-GAN	3DBF-GAN	Spec-GAN	3DBF-GAN
*Overall Accuracy* (%)	56.51	75.62	63.66	77.94	77.17	87.63
*Average Accuracy* (%)	70.04	81.05	72.85	81.36	73.18	92.30
*F1 Score*	60.60	79.10	66.97	77.30	81.09	91.09

**Table 17 sensors-23-08562-t017:** Overview of research works exploring the use of GANs and SGANs in the agrifood field.

Application Domain	Dataset	DA Method	Ref.
F: Generate realistic synthetic samples with GAN to predict oil content of single maize kernel.	NIR (866–1701 nm)	DCGAN to generate samples and balance the dataset	[71]
F: Use GANs to improve the model performance to accurately distinguish cumin and fennel from three different regions.	MIR (1000–2500 nm)	DCGAN to generate samples and balance the dataset	[72]
F: Use a GAN to enhance the dataset to achieve early detection of tomato spotted wilt virus.	VIS-NIR (395–1005 nm)	DCGAN to generate samples and balance the dataset	[73]
F: Enhance a Raman spectra dataset for detection of foodborne pathogenic bacteria.	MIR (2857–25,000 nm)	GAN for Raman spectroscopy	[76]
A: Enhance HSIs dataset to identify and help to preserve fields from weed infestations.	NIR (660 and 790 nm)	SGAN based on DCGAN as generator of HSIs	[87]
A: Enhance a multispectral images dataset to differentiate between fields with crops and weeds.	VIS–NIR (350–1050 nm)	SGAN based on DCGAN as generator of HSIs	[89]
A: Enhance a multispectral images dataset with a bilateral filter.	VIS–NIR (430–860 nm)	DCGAN for hyperspectral images	[91]
B: Development of a rapid identification method for pathogens in water.	MIR (5556–16,667 nm)	DCGAN to enhance the dataset	[75]
MC: Improving the *accuracy* and efficiency of hyperspectral image classification in the field of remote sensing.	VIS–NIR (400–2200 nm)	DCGAN for hyperspectral images	[90]

F: food adulteration; A: agriculture; B: biology; MC: model calibration.

**Table 18 sensors-23-08562-t018:** Comparison of DA methods.

Method	Advantages	Disadvantages
Non-DL-Based Methods	Easy to implement. Quick to apply. Creates semantic variability. Useful for class imbalance, can enhance the minority class representation.	May degrade data quality. May generate unrealistic data. Does not add real semantic variability. Sensitive to noise and outliers.
GAN-Based Methods	Generates realistic data. Adds real semantic variability. Ability to learn complex data distributions.	Time and computational resource consuming. Can be challenging to train. Difficult to evaluate the model’s generator.
SGANs Methods	Utilizes both labeled und unlabeled data. May enhance generation quality. Could improve model generalization. Ability to learn complex data distributions.	More complex to implement and train. Requires time and computational resources. Difficult to evaluate the generator performance.

## Data Availability

Not applicable.

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
