# Peer review of "Data Augmentation Techniques for Machine Learning Applied to Optical Spectroscopy Datasets in Agrifood Applications: A Comprehensive Review"

_sensors, 2023, doi:10.3390/s23208562_

Round 1

Reviewer 1 Report

In this manuscript, a brief overview of different data analysis techniques to optical spectroscopy datasets obtained from agrifood industry real applications. The manuscript is a good reference for readers in the field of researching and processing data for optical spectroscopy in agrifood. However, the manuscript is somewhat disorganized.

1. The structure of the whole manuscript is somewhat disorganized, and this article aims to provide an overview of data augmentation techniques for optical spectroscopy. However, there is little discussion on how spectral techniques can be applied to agrifood. This will cause confusion for readers who are not familiar with optical spectroscopy in agrifood applications. The authors should add a review of optical spectroscopy in agrifood applications in section 2.

2. The authors proposed CM, AUC, ROC, MCC, et al. as the evaluation index in Section 2, and provided a detailed introduction to the calculation method of MCC. However, in the analysis of the performance of different data processing technologies in Section 3, it is rarely found that these indicators were used for discussion. It is recommended that the author provide these parameters. Meanwhile, the authors should provide specific analysis and outlook on how to improve these performance parameters.

3. Please provide a table for discussion, in which some information on the main merits or disadvantages of different data augmentation technologies should be included, so that readers can have a more comprehensive understanding of the characteristics of each enhancement technology.

4. Please pay attention to the “Error! Reference source not found” on lines 224, 232, 239, 248, 649, and 626. And check the entire manuscript.

5. Page 4, line 150, title 1.2.1 should be changed to 2.2.1, and page 5, line 175, title 1.2.2 should be changed to 2.2.2. Please carefully review the entire manuscript.

The quality of English language is good.

Reviewer 2 Report

Dear Authors,

Since data augmentation (DA) techniques are used to improve the performance of machine learning and deep learning techniques, this review study presents the application of DA techniques to optical spectroscopy datasets obtained from real applications of the agri-food industry. The use of simple DA techniques is described, as well as the use of more complex algorithms based on deep learning generative adversarial networks (GANs) and semi-supervised generative adversarial networks (SGANs). The article is a study that will contribute to the literature in general, and it would be appropriate to make the following corrections.
1. It would be appropriate to add a statement about artificial intelligence techniques in the article title.
2. P3-L80: The statement "The first section (section 2)..." misleads the reader. "The first section" should be corrected to "The second section". Other section descriptions should also be arranged in this way.
3. P4-L131: "ec.1" should be corrected to "eq. 1" and all "ec" abbreviations should also be corrected.
4. P4-L153: Must be written as 2 superscripts in the abbreviation "R2". The situation is similar throughout the article and should be corrected.
5. P7:L224: "Error! Reference source not found." The statement should be corrected. This statement is numerous in the article and should be corrected in its entirety.
6. P7:L252: The period at the end of the title should be deleted.
7. P8-L269: The word "gaussian" should be corrected to "Gaussian" in the entire article.
8. P11:L363: The expression "35dB of noise" should be changed to "35 dB".
9. P12:L391: R2 value should be checked.
10. P13:L406:"FAR-IR (2500 - 18,000 nm) statement should be checked.
11. In general, shape resolutions should be increased.

Regards,

Minor editing of English language required

Reviewer 3 Report

Machine learning and deep learning combined with data augmentation techniques are promising tools in spectroscopy and image analysis, especially for biomedical, agricultural and biomedical application, when obtaining large datasets could be challenging. The manuscript presents a comprehensive review on the application of various data augmentation techniques in spectroscopy for the agrifood industry.

In my opinion the paper is well written and well structured, the list of references is adequate to the current state of this field of research and technology. The information presented in the review is relevant, systematic, easily understandable, and useful for readers.

Unfortunately, any previous reviews on this topic were not mentioned in the manuscript. What new contributions in terms of overviewing recent publications or comprehensive and systematical analysis were made by the authors compared to other similar reviews?

Round 2

Reviewer 1 Report

All revisions indicated by the reviewers in the manuscript have been made. 

The quality of English language is relative high.